# Integrated Management of *Chrysodeixis chalcites* Esper (Lepidoptera: Noctuidae) Based on *Trichogramma achaeae* Releases in Commercial Banana Crops in the Canary Islands

**Miguel A. Dionisio** [1,*] and **Francisco J. Calvo** [2]

1 R&D Department, Koppert España, 35018 Las Palmas de Gran Canaria, Spain
2 R&D Department, Koppert España, 04277 Almería, Spain
* Correspondence: migueldiofer@gmail.com

**Abstract:** *Chrysodeixis chalcites* is one of the major pests on banana in the Canary Islands (Spain), where it is widely distributed and causes significant economic losses when larvae feed on developing leaves and fruits. Control of this pest is based on a limited number of pesticides, as growers lack other effective solutions, including biological control. Nevertheless, previous studies have shown some potential against *C. chalcites* of the egg parasitoid *Trichogramma achaeae*. We conducted two field trials over two consecutive years in commercial banana crops, comparing the results against *C. chalcites* of augmentative *T. achaeae* releases (IPM) and conventional chemical control. In total, 215 and 366 wasps/m$^2$ were released in 2016 and 2017, respectively. Naturally occurring *Trichogramma* species contributed to *C. chalcites* control in both treatments, but the maximum number of parasitized eggs in IPM plots nearly tripled and doubled those recorded in chemical control plots in 2016 and 2017. Thus, *T. achaeae* releases significantly increased the parasitism by more than 10% compared to that observed in chemical control plots, amounting to 65.6 ± 7.7% and 56.7 ± 5.8% in 2016 and 2017, respectively. This was enough to keep the pest at tolerable levels in IPM plots, avoiding the need for pesticide applications, whereas in chemical control plots, repeated applications were needed during the experiment for that. Moreover, there were not significant differences in the abundance of larvae and severity of plant damage (<6% in 2016 and <12% in 2017), even when pest pressure in 2016 was two-fold greater in IPM plots. Fruit damage was also similar between treatments in 2016 (<2%) but was significantly reduced from 5.1 ± 1.5% in chemical control to 4.2 ± 1.4% in IPM in 2017. Overall, this study provides guidelines for the biological control of *C. chalcites* in banana, based on augmentative releases of *T. achaeae*, and demonstrates that this method can be effective, constituting an environmentally and sustainable alternative to conventional chemical control for banana growers.

**Keywords:** banana crops; *Trichogramma*; lepidoptera; biological control; integrated pest management

## 1. Introduction

Banana (*Musa acuminata* Colla) is an important crop worldwide, with an annual production over 116 million tons [1]. Around 9000 ha are cultivated in the Canary Islands (Spain), where more than 400,000 t are produced every year, which represents ca. 60% of the total European production [2,3]. The golden twin spot moth *Chrysodeixis chalcites* Esper (Lepidoptera: Noctuidae) is one of the major pests of banana plantations in the Canary Islands [4]. Pest larvae feed preferably on growing unfolding leaves and young fruits, which lose their economic value [5]. Farmers have typically relied on pesticides for *C. chalcites* control in banana as they lacked other effective solutions, including biological control [6]. However, the inadequate use of a limited number of pesticides has led to the generation of resistance by the pest [7,8]. This, together with the increasing limitations on pesticide use by governments, retailers and consumers, due to their negative impact on non-target organisms, the environment and human health, [9–12] is leaving growers without pesticides for caterpillar control in banana. This has led to an increasing interest in

integrated pest management (IPM) strategies that combine biological control agents and synthetic pesticides, thereby providing growers with additional control options [12,13].

There are several naturally occurring predators and parasitoids of *C. chalcites* in banana crops in the Canary Islands [14–16], although they have not been tested sufficiently under realistic crop conditions and/or are not commercially available yet [17,18]. Several species of the genus *Trichogramma* (Hymenoptera: Trichogrammatidae) that are among the species attacking *C. chalcites* in the Canaries have been identified as promising biological control agents of this pest [15]. Some of these parasitic wasps are commercially available in several formulations that facilitate their introduction in open field or greenhouse crops [19–21]. Interestingly, *T. achaeae* Nagaraja and Nagarkatti, which is the most abundant *Trichogramma* species in the Canary Islands [22,23] and has been commercially available in Spain since 2009 [24], is able to reduce plant damage by *C. chalcites* in banana, in combination with applications of *Bacillus thuringiensis* var. *kurstaki* [15]. However, the development of long-term field studies is still necessary in order to provide an adequate strategy for growers [7]. Thus, we conducted two field trials in commercial banana crops from the Canary Islands over two entire consecutive crop seasons to evaluate the results against *C. chalcites* of augmentative *T. achaeae* releases.

## 2. Materials and Methods

### 2.1. Experimental Sites and Design

The study consisted of two field experiments, of which the first was conducted in 2016 and the second in 2017, both coinciding with the typical growing seasons of banana crops in the Canary Islands. In each trial, three pairs of commercial banana greenhouses, each pair located on a different farm, were chosen for the experiments. In both experiments, all farms had records of *C. chalcites* in previous years and were located south of Tenerife, which is the geographic area most affected by *C. chalcites* within the Canary Islands. Pairs belonging to the same farm had the same characteristics (crop age, cultivar, structure, orientation, management, cropping system, etc.) and were randomly assigned to the two compared treatments: (1) chemical control or (2) biocontrol-based IPM (IPM) (Table 1). Treatments were then compared in a complete randomized block design with three replicates of two treatments. Plots belonging to the same replicate were separated by ca. 50 m to avoid the immigration into chemical plots of *T. achaeae* released in IPM plots.

**Table 1.** Characteristics of the experimental sites.

| | 2016 Trials | | | 2017 Trials | | |
|---|---|---|---|---|---|---|
| | **Replicate 1** | **Replicate 2** | **Replicate 3** | **Replicate 1** | **Replicate 2** | **Replicate 3** |
| Municipality | Arona | San Miguel | Adeje | Arona | Arona | San Miguel |
| Latitude | 28°1′25.00″ N | 28°2′15.19″ N | 28°6′7.27″ N | 28°1′25.00″ N | 28°2′15.81″ N | 28°3′4.32″ N |
| Longitude | 16°40′45.08″ W | 16°37′54.13″ W | 16°44′50.31″ W | 16°40′45.08″ W | 16°38′58.14″ W | 16°37′55.84″ W |
| Altitude | 40 MASL | 99 MASL | 31 MASL | 40 MASL | 85 MASL | 158 MASL |
| Crop Age | >10 years | >10 years | >10 years | >10 years | >10 years | >10 years |
| Variety | Grand Nain | Gruesa Palmera | Gruesa Palmera | Grand Nain | Gruesa Palmera | Grand Nain |
| Chemical C | | | | | | |
| Size (m²) | 1895 | 11,400 | 4600 | 1895 | 5850 | 2000 |
| Plants | 379 | 2111 | 767 | 379 | 936 | 360 |
| IPM | | | | | | |
| Size (m²) | 1300 | 5700 | 4000 | 1300 | 3750 | 3000 |
| Plants | 260 | 1056 | 733 | 260 | 600 | 540 |
| Total Size | 3195 | 17,100 | 9000 | 3195 | 9600 | 5000 |
| Total Plants | 639 | 3167 | 1500 | 639 | 1536 | 900 |

Chemical C.: chemical control management; IPM: biological-based integrated pest management; MASL: meters above sea level.

Management of *C. chalcites* in chemical control plots was based exclusively on the use of pesticides. Pesticides were sprayed when the mined leaf area exceeded 10% and/or

when new larval damage was detected on fruits [25] (Table 2). All pesticide applications in 2016 and 2017 experiments are shown in Tables 3 and 4, respectively.

**Table 2.** Criteria to establish the requirement and rates for parasitoid releases in 2016.

| No Trap Catches in Pheromone Trap | | Trap Catches in Pheromone Traps > 0 | |
| --- | --- | --- | --- |
| No new pest damage & <10% defoliation | 10 wasps/m$^2$ × week | No new pest damage & <10% defoliation | 75 wasps/m$^2$ × week |
| <10% defoliation with larvae on crop | 75 wasps/m$^2$ × week | <10% defoliation with larvae on crop | 75 wasps/m$^2$ × week |
| ≥10% defoliation with larvae on crop | Selective pesticide | ≥10% defoliation with larvae on crop | Selective pesticide |
| Fresh fruit damage and/or larvae on fruits | Selective pesticide | Fresh fruit damage and/or larvae on fruits | Selective pesticide |

**Table 3.** Timing and expected and real (between brackets) rates for *T. achaeae* releases (insects/m$^2$) (i/m$^2$) and insecticide applications in each plot in 2016 trials.

| Dates | Replicate 1 | | Replicate 2 | | Replicate 3 | |
| --- | --- | --- | --- | --- | --- | --- |
| | IPM | Chemical C. | IPM | Chemical C. | IPM | Chemical C. |
| 08 June | 75 i/m$^2$ | IND | 75 i/m$^2$ | IND | 75 i/m$^2$ | |
| 22 June | 75 i/m$^2$ | CLP | 75 i/m$^2$ | | 75 i/m$^2$ | |
| 06 July | | | 75 i/m$^2$ | | | |
| 20 July | | | 75 i/m$^2$ | | | |
| 03 August | | | 75 i/m$^2$ | CLP | 75 i/m$^2$ | |
| 17 August | 75 i/m$^2$ | | 75 i/m$^2$ | | 75 i/m$^2$ | |
| 31 August | 75 i/m$^2$ | | | | 75 i/m$^2$ | |
| 14 September | 75 i/m$^2$ | | | | | |
| 28 September | 75 i/m$^2$ | | 75 i/m$^2$ | | 75 i/m$^2$ | |
| 12 October | 75 i/m$^2$ | | | | | |
| 26 October | 75 i/m$^2$ | | 75 i/m$^2$ | | 75 i/m$^2$ | |
| 09 November | 75 i/m$^2$ | | 75 i/m$^2$ | | 75 i/m$^2$ | |
| 23 November | 75 i/m$^2$ | | 75 i/m$^2$ | | 75 i/m$^2$ | |
| Total | 750 i/m$^2$ (258 i/m$^2$) | | 750 i/m$^2$ (195 i/m$^2$) | | 675 i/m$^2$ (193 i/m$^2$) | |

IPM: biological-based integrated pest management. Chemical C.: chemical control management; CLP: chlorpyrifos 48% (DURSBAN® Aventis); IND: indoxacarb 30% (STEWARD® DuPont).

**Table 4.** Insecticide applications in each plot in 2017.

| Dates | Replicate 1 | | Replicate 2 | | Replicate 3 | |
| --- | --- | --- | --- | --- | --- | --- |
| | IPM | Chemical C. | IPM | Chemical C. | IPM | Chemical C. |
| 24 May | | | | | *Bt* | *Bt* |
| 21 June | | | | | | |
| 19 July | | | | | *Bt* | *Bt*, IND |
| 27 September | | | *Bt* | *Bt* | | |
| 11 October | | IND | | | | |
| 08 November | | CLP | | | | |

IPM: biological-based integrated pest management. Chemical C.: chemical control management; *Bt*: *Bacillus thuringiensis* var. *kurstaki*; CLP: chlorpyrifos 48% (DURSBAN® Aventis); IND: indoxacarb 30% (STEWARD® DuPont).

In IPM plots, management of *C. chalcites* was based on augmentative releases of *T. achaeae* and supplementary pesticide applications with *B. thuringiensis*. The parasitoid was obtained from the commercial product TRICHOGRAMMA ACHAEAE® (Koppert, Berkel en Rodenrijs, The Netherlands), which consisted of 6 × 8 × 0.4 cm cardboard cards containing parasitized *Ephestia kuehniella* Zeller (Lepidoptera: Pyralidae) eggs of different stages from which ca. 2500 *T. achaeae* adults were delivered into the crop over a two-week

period. Cards were initially hung from plants using the integral hook, but this did not prevent ant predation on parasitized eggs (see sampling). Cards were then placed (from 01/08 onwards in 2016 and for the entire experiment in 2017) inside $5 \times 5 \times 6$ cm cardboard boxes D-BOX® (Koppert, Berkel en Rodenrijs, The Netherlands) that were hung from the plants using a $20 \times 5$ cm yellow sticky band (ROLLER TRAP MINI®, Koppert, Berkel en Rodenrijs, The Netherlands) that served as a barrier for the ants. In all cases, cards or D-Boxes containing the cards were always evenly distributed throughout the plot and placed within the plant canopy to protect them from direct sunlight.

Adult catches in pheromone traps, presence of larvae on plants, and plant and fruit damage (Table 2) were used to establish the requirement, timing and rates for parasitoid releases in 2016 (Table 3). This release schedule was adapted from previous field trials releasing *Trichogramma* spp. parasitoids against *C. chalcites* and other pests [15,26–28] and taking into account the dynamics of *C. chalcites* populations in the Canary Islands [29]. In 2017, based on the results of 2016, this release schedule was shifted to a program including two rounds of releases. This was done to simplify the strategy and to make it economically viable. The first round began the first week of March and consisted of four releases every two weeks. The second round was initiated in the last week of May and finished at the end of August after eight releases, conducted every two weeks. The expected rate for all releases in 2017 was 50 wasps/m$^2$. Rates of ant predation on eggs inside cards and the emergence of adults from cards (see sampling section) were used to estimate the real quantity of adults released into experimental plots by correcting the expected rate of release with these rates. Predation by ants was 38% and 0% in 2016 and 2017, respectively, and 66% and 61% of adults emerged from parasitized eggs in 2016 and 2017. Thus, 366 adults/m$^2$ were released in total in plots receiving *T. achaeae* releases in 2017. In 2016, 215 adults/m$^2$ were released, on average, in IPM plots, although the timing and expected and real rates for all releases are shown in Table 3. *B. thuringiensis* was sprayed in 2017 only, and applications are shown in Table 4.

### 2.2. Ambient Conditions

Temperature and relative humidity were monitored in each replicate by using a LogTag HAXO-8 Data Logger (HOBO) placed within the plant canopy. An average daily temperature of $23.9 \pm 0.14$ °C (max: $32.6 \pm 0.18$ °C; min: $18.6 \pm 0.13$ °C) and $22.4 \pm 0.18$ °C (max: $28.5 \pm 0.26$ °C; min: $18.5 \pm 0.16$ °C) were recorded in 2016 and 2017, respectively. Average relative humidity during the experiment was $67.9 \pm 0.32\%$ (max: $83.0 \pm 0.34\%$; min: $45.2 \pm 0.44\%$) and $73.8 \pm 0.37\%$ (max: $84.3 \pm 0.27\%$; min: $58.8 \pm 0.49\%$) in 2016 and 2017, respectively.

### 2.3. Sampling

All plots were inspected every two weeks for 17 weeks in 2016, from 11 May to 21 December, and for 22 weeks in 2017, from 01 March to 20 December. In 2016, populations of *C. chalcites* and their natural enemies were recorded on seven randomly selected plants, four plants and three suckers and seven randomly selected clusters in each plot. The number of parasitized and non-parasitized eggs, $L_1$–$L_2$ (length < 1cm) and $L_3$–$L_5$ (length $\geq$ 1cm) larvae of *C. chalcites* were counted on the five youngest leaves selected from the upper part of each of the selected plants and on the five upper rows of fruits of each selected cluster. Plant and fruit (whole cluster including petiole) damage was recorded by visually rating the mined area by *C. chalcites* as 0, 1, 2, 3 or 4 when the mined area was 0, 5–20%, 21–40%, 41–60% or >60% respectively [15]. In 2017, mother plants were excluded from sampling, as data from 2016 demonstrated that observations from daughter plants were representative of the whole population of the pest and natural enemies in the crop. The same parameters as in 2016 were recorded in each of the three youngest leaves selected on each of the fifteen plants and fifteen clusters randomly selected in each plot. Additionally, in 2016, four pheromone traps LEPISAN®, (Koppert, Berkel en Rodenrijs, The Netherlands) were placed in all plots to record the flying activity of adults, to serve as an additional

criterion to establish the need for *T. achaeae* releases or pesticide applications. Pheromone dispensers were replaced every four weeks.

The species complex parasitizing *C. chalcites* eggs in each plot was studied by collecting parasitized and non-parasitized *C. chalcites* eggs during each sampling (n ≥ 30 eggs/plot). They were packed in separate plastic containers by plot and brought to the laboratory, where they were kept at room temperature until adult parasitoids emerged or a *C. chalcites* larva hatched. Adult parasitoids were then mounted on microscope slides and identified as species using a compound microscope [22].

Ant predation on *T. achaeae* cards was assessed throughout the trials by taking fifteen randomly selected cards in each plot. Cards were taken four weeks after the release, at which time they were opened and visually rated as 0, 1, 2, 3 or 4 when the predated area was 0, <25%, 26–50%, 51–75%, >76% respectively.

Additionally, the percentage of emergence in the field, i.e., the percentage of adults emerging from parasitized eggs in experimental plots, was estimated. For that, four cards of each release were placed in a 100 mL ventilated plastic container. Cards were kept for four weeks inside the crop, at which time the containers were brought to the lab and the emerged wasps counted using a stereomicroscope.

### 2.4. Data Analysis

Treatment effects on pest levels, parasitism rates and fruit and plant injury were analyzed using generalized linear mixed effects models GLMM ($\alpha = 0.05$), with time (weeks) as the random factor nested in blocks (replicates) to correct for pseudoreplication due to repeated measures [30,31]. Insect numbers per leaf and percentage of *C. chalcites* parasitism were log (x + 1) and arcsin $\sqrt{(x)}$ transformed, respectively, prior to analysis to stabilize error variance, although untransformed values are given in the text. The percentages of plant and fruit damage were estimated from rates using the Townsend and Heubergur formula [32]:

$$\text{Damage } \% = \sum \left( \frac{n \cdot v}{N \cdot V} \right) \times 100$$

where *n* is the number of sampling units in each category, *v* is the value for each category, *N* is the total number of sampling units, and *V* is the value of each highest category.

### 3. Results

#### 3.1. Population Dynamics of Chrysodeixis chalcites

Although in 2017, the number of non-parasitized *C. chalcites* eggs was, on average, at higher levels than in 2016, the dynamics followed a similar pattern in both years in all plots. In both years and treatments, levels of non-parasitized eggs of *C. chalcites* were close to 0 at the beginning of the experiments and increased progressively from the beginning of May until the middle of October, when they declined rapidly until December when the experiments concluded (Figure 1A,C). The number of non-parasitized eggs per leaf was, on average, at similar numbers in plots belonging to both tested treatments within the same year, and those were not significantly different (2016: $F_{1,118} = 0.992$; $p = 0.321$; 2017: $F_{1,130} = 1.547$; $p = 0.216$).

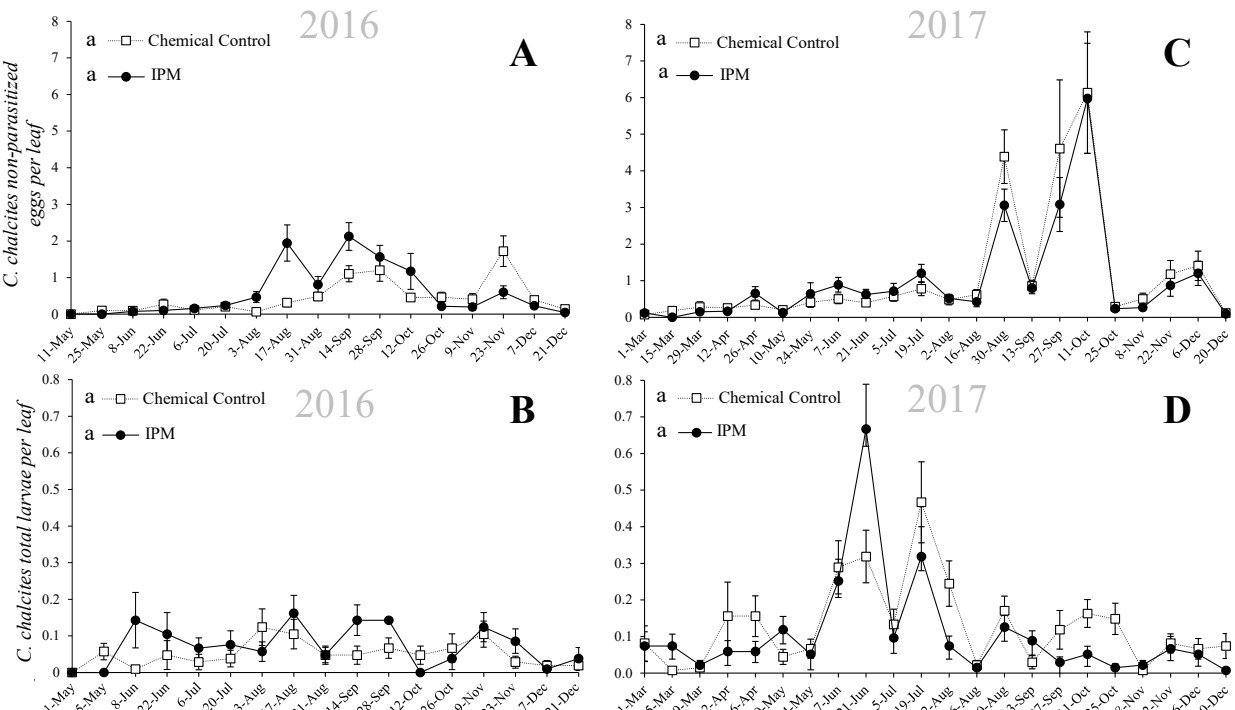

**Figure 1.** Mean (±SE) of total *C. chalcites* eggs per leaf in 2016 (**A**) and 2017 (**C**) and total larvae per leaf (mean ± SE) in 2016 (**B**) and 2017 (**D**) in each treatment during the experiments. Legends with different letters were significantly different (GLMM, *p* < 0.05).

Similarly to eggs, the numbers of larvae per leaf were low at the beginning of the experiment and followed a similar pattern in plots belonging to both tested treatments in both 2016 (Figure 1B, $F_{1,118}$ = 0.007; *p* = 0.935) and 2017 (Figure 1D, $F_{1,130}$ = 1.167; *p* = 0.282) and therefore were not significantly different. In 2016, the number of larvae per leaf remained nearly constant during the experiment and never exceeded 0.2 in any plot. In 2017, the abundance of larvae peaked several times during the experiment and reached ca. 0.7 per leaf in IPM plots in June, a bit higher than the maximum of ca. 0.5 per leaf recorded in chemical control plots.

### 3.2. Incidence of Parasitism

Virtually no parasitized eggs were observed in any treatments at the beginning of both trials. The dynamics of parasitized eggs was similar to that observed for non-parasitized ones in both years (Figure 1A,C), with a progressive increase from the spring until September–November, when the highest values were observed. The maximum number of parasitized eggs in IPM plots nearly tripled and doubled those recorded in chemical control plots in 2016 and 2017, respectively. On average, more parasitized eggs were counted in most of the weeks in plots receiving *T. achaeae* in both years, and consequently the abundance of parasitized eggs was higher in IPM compared to chemical control plots (2016: Figure 2A, $F_{1,118}$ = 10.139; *p* = 0.002; 2017: Figure 2C, $F_{1,130}$ = 8.395; *p* = 0.004).

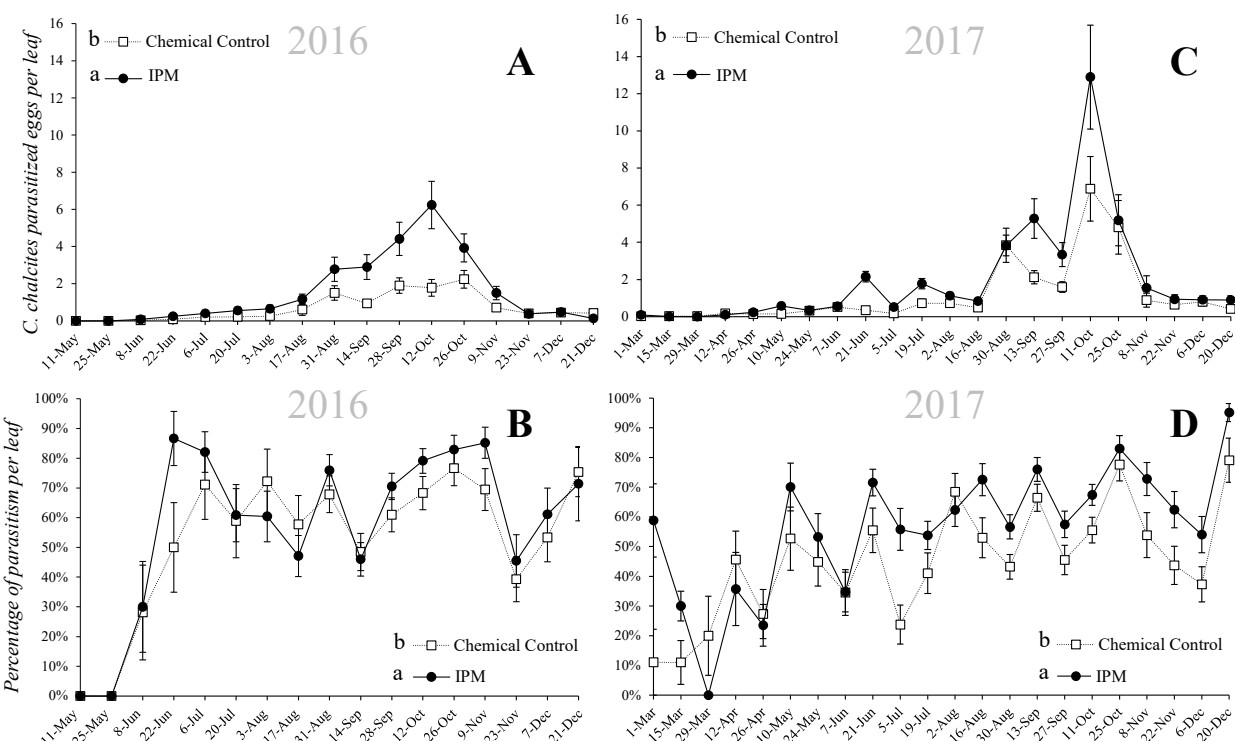

**Figure 2.** Number (mean ± SE) of parasitized *C. chalcites* eggs per leaf in 2016 (**A**) and 2017 (**C**) and incidence of parasitized (percentage ± SE) *C. chalcites* eggs per leaf in 2016 (**B**) and 2017 (**D**) in each treatment. Legends with different letters were significantly different (GLMM, *p* < 0.05).

The percentage of parasitized eggs was also significantly higher in IPM than in chemical control plots in both years (2016: Figure 2B, $F_{1,103}$ = 4.638; *p* = 0.034; 2017: Figure 2D, $F_{1,121}$ = 30.044; *p* < 0.001); despite the dynamics being similar in plots belonging to both tested treatments, it was higher in most of the weeks in plots with parasitoid releases. In both years, in IPM plots, the percentage of parasitism peaked above 90% in some weeks, although in 2016, it increased more rapidly over the first weeks of the experiment and later varied from values between 50 to 90%, whereas in 2017 it increased progressively throughout the experimental period.

### 3.3. Leaf and Fruit Damage

Damage of *C. chalcites* to leaves was similar in all plots during both years (Figure 3A,C), and thus there were no significant differences between chemical control and IPM plots in the intensity of plant damage by *C. chalcites* (2016: $F_{1,117}$ = 2.069; *p* = 0.153; 2017: $F_{1,130}$ = 0.206; *p* = 0.651). Two peaks of leaf damage were observed during the experiment in both years, the first (highest) between the end of May and the beginning of August and the second between the end of August and the beginning of October. Nevertheless, the highest percentage of leaf damage was 5.95 ± 1.81% and 5.48 ± 1.33% in 2016 and 11.11 ± 1.14% and 12.04 ± 1.17% in 2017 in chemical control and IPM plots, respectively.

Fruit damage was not significantly different between treatments in 2016 ($F_{1,118}$ = 0.006; *p* = 0.939) as similar percentages of mined fruit area were observed in all plots (Figure 3B), but in 2017 fruit damage was significantly more severe in chemical control than in IPM plots ($F_{1,130}$ = 5.398; *p* = 0.022) (Figure 3D). In 2016, the mined area on fruits decreased during the year after a rapid increase at the beginning of the season, whereas in 2017, it remained nearly constant, varying from ca. 2% to 6% throughout the season.

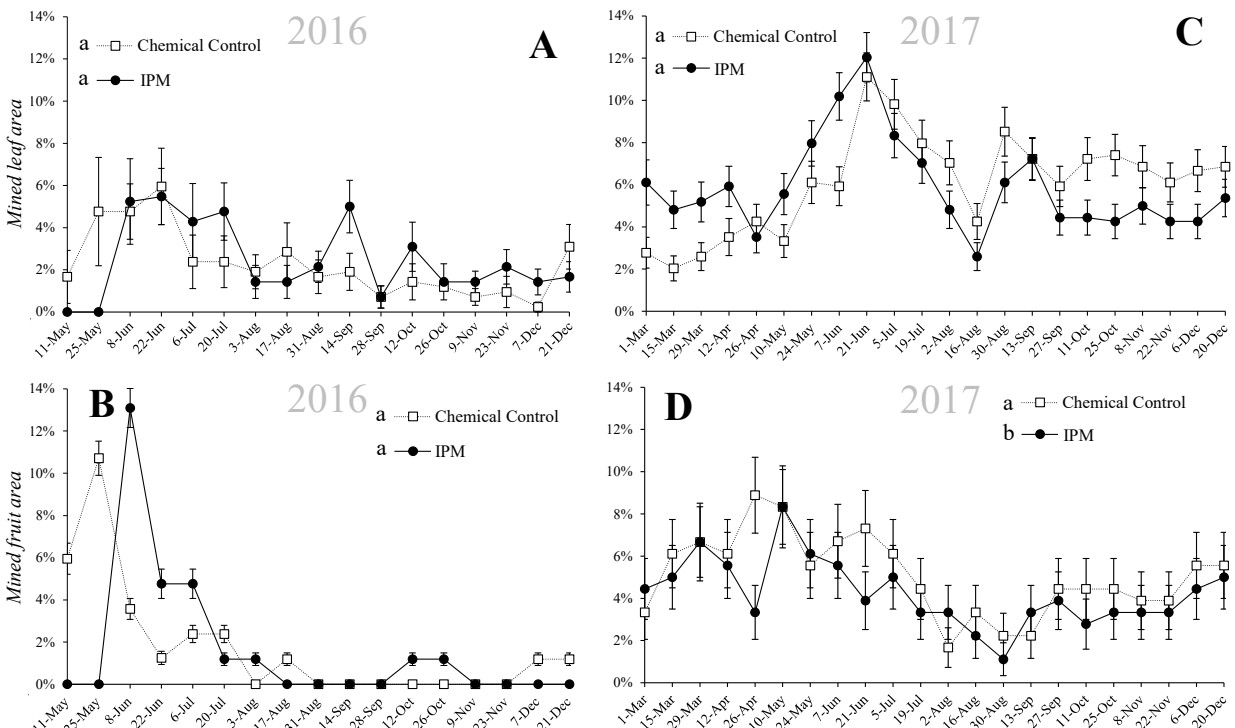

**Figure 3.** Percentage (mean ± SE) of leaf damage in 2016 (**A**) and 2017 (**C**) and percentage of fruit damage (mean ± SE) in 2016 (**B**) and 2017 (**D**) caused by *C. chalcites* larvae in each treatment. Legends with different letters were significantly different (GLMM, *p* < 0.05).

*3.4. Relative Abundance of Egg Parasitoids*

All adult parasitoids emerging from *C. chalcites* eggs taken from leaf samples in 2016 and 2017 belonged to the *Trichogramma* genera. *Trichogramma achaeae* was the most abundant species in plots with parasitoid release, constituting 74.8 ± 7.0% in 2016 and 67.1 ± 4.5% in 2017 of adults emerging from field samples. In chemical control plots, the incidence of *T. achaeae* amounted to 29.6 ± 6.0% and 33.4 ± 10.8% of adults emerging from egg samples taken in 2016 and 2017, respectively.

## 4. Discussion

In our study, despite pest densities being similar between plots at the beginning of both seasons, it was observed that IPM plots were under a much higher pest pressure than chemical control plots when parasitized and non-parasitized eggs were added. This might be explained because some chemical pesticides can have a repellent effect on pests [33] and thus fewer eggs were laid on plants belonging to chemical control than on those located in IPM plots. In any case, our study demonstrated that the successful biological control of *C. chalcites* in commercial banana plantations in the Canary Islands following fortnightly releases of the egg parasitoid *T. achaeae* is possible. This is consistent with earlier results that suggested that *T. achaeae* could be a promising biological control agent for *C. chalcites* in this crop [7,15,28]. These authors reported good *C. chalcites* control in banana by combining *T. achaeae* releases and applications of *B. thuringiensis* but concluded that more studies were necessary to establish an appropriate strategy. This study offers a suitable tactic for the successful control of *C. chalcites* in banana, based almost exclusively on *T. achaeae* releases, which complements these earlier studies.

In our trials, we recorded ca. 25-fold greater levels of *C. chalcites* eggs than larvae in both seasons. This suggests that pest populations were mostly regulated by biotic and abiotic factors affecting eggs. This could explain why parasitoid releases in IPM plots, which significantly increased parasitism with respect to chemical plots, provided good and even better results against *C. chalcites* than chemical control. This increased parasitism in

IPM plots could have been in response to the *T. achaeae* releases that not only increased the abundance of parasitic wasps in these plots but also served to increase the relative abundance of the best-adapted *Trichogramma* species to the local conditions and/or to attack and parasitize *C. chalcites*. For instance, *T. achaeae* developed and reproduced well between 15 to 30 °C, with an optimal temperature for development at 25 °C and for reproduction from 20 to 25 °C [9]. Thus, the thermal regime present in the Canary Islands, and the one we observed in our study, is close to optimal for *T. achaeae* and can explain why it is the most frequent and widely distributed *Trichogramma* species in the archipelago [15,17] and in our trials. The parasitism observed in chemical control plots could be partly due to *T. achaeae* adults immigrating from IPM plots [34,35]. Nevertheless, the distance between IPM and chemical control plots and the fact that del Pino, 2022 [15] found similar levels of parasitism in plots with no parasitoid releases, might suggest that most of the parasitism in chemical control plots was due to natural occurring parasitoids. Additionally, the differences that we have observed in our studies between IPM and chemical control plots are consistent with earlier studies [15].

The option of controlling *C. chalcites* with *T. achaeae* following the schedule we developed in this study offers different technical advantages: (1) simpler control strategy; (2) healthier and more environmentally friendly control strategy; (3) lower risk of the development of pest resistance; and (4) being an economical and ready-to-use solution. Releasing the parasitoid every two weeks, coinciding with the period in which the pest is more problematic, and always at the same rate simplifies the control strategy technically. The need for *T. achaeae* releases does not rely on captures in traps or direct field observations (sampling), thus reducing scouting efforts and facilitating decision making.

The pest could be controlled without spraying synthetic pesticides; thus, the fruits would be free of pesticide residues. This constitutes a perfect answer for growers to the increasing limitations on pesticide use and customers' and retailers' demands for healthier products [13,36,37]. Moreover, the reduction in the use of wide-spectrum and non-selective pesticides would benefit the control of *C. chalcites* among other pests, as it would decrease the negative impact on beneficial insects, facilitating their immigration into the crop. The occurrence of the larval parasitoid *Cotesia* sp. (Hymenoptera: Braconidae), predators such as *Chrysoperla carnea* (Neuroptera: Chrysopidae), *Cyrtophora citricola* or *Neoscona crucifera* (Araneae: Araneidae), which are common in the Canaries [14–16,38], could complement the effects of *T. achaeae* releases on *C. chalcites* populations. For instance, numerous studies have shown that *B. thuringiensis* is safe for *Trichogramma* species [39–41]. Contrarily, chlorpyriphos is very harmful for *Trichogramma* species [41,42] and other biological control agents [43–45], being the most lethal among twenty-two pesticides tested on *T. achaeae* [41]. Indoxacarb, one of the pesticides sprayed in chemical control plots during the experiment, is considered to be moderate harmful for beneficial insects in general [46–49] and relatively safe for *T. achaeae* [7,17,50]. The selection of the right pesticides is quite important to mitigate negative effects not only on the released beneficial insects but also on those that can immigrate naturally into the crop.

The use of *T. achaeae* adds extra options to the toolkit for *C. chalcites* control, so now growers do not have to rely exclusively on the use of pesticides, which might help reduce the number of pesticide applications. This is an ideal way to help avoid the generation of resistance by the pest [8], which is particularly interesting for banana cultivation in the Canaries, where growers have a very limited list of pesticides. Actually, the generation of resistance has already happened in the case of indoxacarb on several other noctuidae pests such as *Helicoverpa armigera* and *H. assulta* (Lep.: Noctuidae) [51,52] or *Spodoptera litura* [53].

The control of *C. chalcites* with *T. achaeae* could be economically feasible, as the total rate of 366 i/m$^2$ for *T. achaeae* releases in our study of 2017 was within the range of recommended rates for other authors in banana and other crops [54]. Del Pino et al. 2010 [28] proposed a total of 425 *T. achaeae*/m$^2$ in combination with five applications of *B. thuringienesis* for *C. chalcites* control in banana. Vila et al. 2010 [27] released 175 i/m$^2$ for *Tuta absoluta* control in tomato, but Cabello et al. 2009 [26] and Desneux et al. 2010 [55] reported that a total

of 525 i/m$^2$ and 1000 i/m$^2$ *T. achaeae* were needed, respectively, to significantly reduce *T. absoluta* damage in tomato. Similarly, Keçeci and Öztop, 2017, [56] had to release 900 i/m$^2$ to control *T. absoluta* in tomato with *T. evanescens*. *T. achaeae* is commercially available in several formulations, some of which are perfectly adapted to banana crops. It is important to mention that the formulation to be used should protect parasitized eggs from ant attack. Otherwise, growers should implement a method similar to that we used in the present study, although this would be more time-consuming. Therefore, the commercial availability of the parasitoid, together with the fact that it is an indigenous parasitoid to the Canary Islands, makes *T. achaeae* a biological control agent that banana growers can start using immediately.

## 5. Conclusions

Overall, our results suggest that increased natural parasitism by *T. achaeae* releases provides effective *C. chalcites* control in banana, thereby reducing growers' dependency on a limited range of pesticides, providing a workable solution for their needs and the market's demands for healthier and safer agricultural products and the protection of the environment. Therefore, *T. achaeae* releases can also contribute to environmental sustainability, biodiversity conservation and the improvement of human health.

**Author Contributions:** Conceptualization, F.J.C.; methodology, F.J.C.; software, F.J.C. and M.A.D.; validation, F.J.C. and M.A.D.; formal analysis, F.J.C.; investigation, M.A.D.; resources, F.J.C.; data curation, F.J.C. and M.A.D.; writing—original draft preparation M.A.D.; writing—review and editing, F.J.C. and M.A.D.; visualization, F.J.C.; supervision, F.J.C.; project administration, F.J.C.; funding acquisition, F.J.C. and M.A.D. All authors have read and agreed to the published version of the manuscript.

**Funding:** This research was supported by Koppert España S.L. and the Ministry of Science and Innovation (Industrial Doctorate Program DI-15-07788).

**Acknowledgments:** We are grateful to all of the farmers who made this study possible by allowing us to conduct field research in their banana plantations. Thanks to Jose Eduardo Belda and Valter Ceppi from Koppert Spain for their confidence in the development of this project and to the Koppert Canarias team for their help in logistics and in the search for farms, Rafael Alonso, Rudy Llarena, Francisco González, Roberto Guillén and Valentina Suarez. Special thanks to Karen Girard (R&D, Koppert Biological Systems, UK) for reviewing the English of the manuscript and all the students and friends who assisted in field and laboratory tasks, Yeray Sagredo, Samuel Cruz and Nauzet Morales.

**Conflicts of Interest:** The authors declare no conflict of interest.

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
