# Peer review of "Integrated Management of Chrysodeixis chalcites Esper (Lepidoptera: Noctuidae) Based on Trichogramma achaeae Releases in Commercial Banana Crops in the Canary Islands"

_agronomy, doi:10.3390/agronomy12122982_

Round 1

Reviewer 1 Report

Thanks to the authors for providing this study.

This study provides practical information on: “Integrated Management of Chrysodeixis chalcites Esper (Lepidoptera: Noctuidae) based on Trichogramma achaeae releases in commercial banana crops in the Canary Islands”.

Mian comment: The study is very good and useful for both researchers in the field and farmers, and it constitutes a sustainable future vision for controlling the studied insect on bananas. It also offers practical solutions to preserve the environment. However, please answer the following comments:

In Abstract:

·       Add to the abstract some important results numbers or percentages.

keywords: Add “banana crops” to keywords.

In the introduction, paraphrase lines 57-59 so that the objective of the study is clearly written.

In material and methods:

·       Where are the specifications and data of the experiment site? Such as latitude and longitude, altitude above sea level, rainfall rate and soil specifications and type. Add that to the study.

·       Put the weather data in the Ambient conditions section into a table or figure to make it clearer.

In Discussion:

·       Put what is written in lines 347-352 in a separate paragraph, which is the conclusion, because it is not found in the study. With the addition of the results of other important results to the conclusion.

Author Response

Reviewer #1

Main comment: The study is very good and useful for both researchers in the field and farmers, and it constitutes a sustainable future vision for controlling the studied insect on bananas. It also offers practical solutions to preserve the environment. However, please answer the following comments:

In Abstract:

  • Add to the abstract some important results numbers or percentages. - Done

keywords:

Add “banana crops” to keywords – Done

In the introduction,

Paraphrase lines 57-59 so that the objective of the study is clearly written - Done

In material and methods:

  • Where are the specifications and data of the experiment site? Such as latitude and longitude, altitude above sea level, rainfall rate and soil specifications and type. Add that to the study.- A new table including all this information has been included.
  • Put the weather data in the Ambient conditions section into a table or figure to make it clearer. It has been provided as a supplementary file as indicated by reviewer #2, we found it as a very good way to provide the information while keeping the reading of the text much simpler and clearer.

In Discussion:

  • Put what is written in lines 347-352 in a separate paragraph, which is the conclusion, because it is not found in the study. With the addition of other important results to the conclusion. Done. We just added this paragraph as conclusion, as we think it summarize the most important results and future directions.

Reviewer 2 Report

The manuscript title “Integrated Management of Chrysodeixis chalcites Esper (Lepidoptera: Noctuidae) based on Trichogramma achaeae releases in commercial banana crops in the Canary Islands” is conducted well and have scientific worth. I have some suggestions for author to improve the current manuscript: 

Reviewer Comments:

1-      In the abstract, the author didn’t mention give any results/data. Just write this “Natural occurring Trichogramma species contributed to C. chalcites control in both treatments…..,”. The authors should mention in the abstract, how much control? How much C. chalcites egg/ larvae reduced? How many mined leaves reduced/ decreased?

2-      In the material and methods section please add the environmental condition data (as a supplementary file), because the experiment is conducted in field and temperature, rain, humidity effect the insect reproduction etc.

3-      In the discussion section please mention the cost of pesticide application and biological control of C. chalcites. Few lines are ok, to show the cost different and effectiveness of biological control of C. chalcites.

4-      Conclusion section is missing in the manuscript. The author should conclude their results and give future direction accordingly.

Author Response

Reviewer #2

Reviewer Comments:

1-      In the abstract, the author didn’t mention give any results/data. Just write this “Natural occurring Trichogramma species contributed to C. chalcites control in both treatments…..,”. The authors should mention in the abstract, how much control? How much C. chalcites egg/ larvae reduced? How many mined leaves reduced/ decreased? - Done

2-      In the material and methods section please add the environmental condition data (as a supplementary file), because the experiment is conducted in field and temperature, rain, humidity effect the insect reproduction etc. It has been provided as a supplementary file.

3-      In the discussion section please mention the cost of pesticide application and biological control of C. chalcites. Few lines are ok, to show the cost different and effectiveness of biological control of C. chalcites. Although we do think this is a very important remark, we decided not to include this information in the text, even in the original version, due to costs can vary quite significantly between geographic areas. Note that the cost of pesticides and natural enemies, as well as labour cost and machinery for their application are very different among Europe, Asia, Africa,… and growers might not have access to the same pesticides due to governmental or market restrictions. However, we provide all the information to readers about the quantities of released insects and the pesticides and the number of applications, so they can easily estimate the costs in their local markets, having the possibility to compare the costs of both control methods.

4-      Conclusion section is missing in the manuscript. The author should conclude their results and give future direction accordingly. We put what was written in lines 347-352 (previous version) as conclusion as suggested by reviewer #1.

Reviewer 3 Report

See only a few corrections in the text

Author Response

Reviewer #3

Provided remarks in separate PDF file and all remarks have been addressed.
